# Logistics Service Supply Chain Vertical Integration Decisions under Service Efficiency Competition

**Xiaomeng Zhang** [1], **Qilan Zhao** [1,*], **Jianjun Zhang** [2] **and Xiongping Yue** [3]

1  School of Economics and Management, Beijing Jiaotong University, Beijing 100044, China
2  College of Economics and Management, Inner Mongolia Agricultural University, Hohhot 010010, China
3  School of Management, Wuhan Polytechnic University, Wuhan 430048, China
*  Correspondence: qlzhao@bjtu.edu.cn

**Abstract:** In the logistics sector, price competition is no longer the only form of horizontal competition between logistics service integrators; instead, it frequently takes the form of service efficiency competition among chains. Facing fierce market competition, vertical resource integration gradually becomes the trend in logistics industry integration. Using the inverse derivation method and comparative analysis, this study examines the relationship between the overall profit of its chain and that of the rival chain under service efficiency competition with or without the integration strategy. Furthermore, it builds two parallel competition logistics service supply chain models based on the inter-chain Nash competition and Stackelberg game of the chain members. The study results demonstrate that when the cost per unit of service efficiency is fixed, the greater the intensity of competition between chains, the more managers should tend to choose an integration strategy to maximize their profits. More interestingly, we find that the optimal integration decision of the supply chain is independent of the competitive intensity when the cost required to improve the unit service efficiency is extremely high.

**Keywords:** competition in services; competitive supply chains; logistics service supply chains; integration decisions

## 1. Introduction

There are 17 sustainable development goals (SDG) defined by the United Nations 2030 Agenda for Sustainable Development [1]. In a context in which globalization is causing both the development and management of operations related to products and services to become more complex [2], stakeholders are increasingly demanding and aware of sustainability challenges along the supply chain [3]. As a result, the supply chain must be handled effectively to meet the requirements of supply chain member enterprises. Vertical integration strategies usually integrate strategic resources and key links in a company's value chain and industrial chain. Vertical integration can achieve rational use of resources, reduce waste, and increase profits. Vertical integration is more diversified in the realization mode. The major methods for achieving vertical integration, according to most academics, are an investment in self-construction, mergers, holding, purchases, and equity participation. Some logistics service firms have vertically integrated with logistics service providers (LSPs) to further boost their core competitiveness and be able to develop sustainably as a result of the rapid development of the logistics service economy and the intensifying competition among logistics enterprises [4]. It is typical for a logistics service integrator (LSI) to team up with a logistics service provider (LSP) in the logistics sector. For example, in maritime logistics service enterprises, members of different segments, including logistics service integrators (shipping companies) and logistics service providers (port operators), establishing vertical alliances to jointly develop and invest in an integrated logistics project will help to progressively create a large-scale logistics supply chain system that is centered on logistics service integrators (LSIs) [5]. The logistics supply chain system has gradually

been formed on a large scale and centered on logistics service integrators. To obtain access to greater market resources, The Maersk Group, the largest shipping corporation in the world, has boosted its investment in ports all around the world. In particular, there is more investment cooperation with Chinese ports. The Maersk Group currently has 10 ports in China, which make up about 25% of all of its ports worldwide. Duffy Group, the world's third-largest French shipping company, is gaining access to international port terminals through a partnership with Shanghai International Port Group. French shipping firm Duffy Group and China Shanghai International Port (Group) Company Limited staged a signing ceremony for a refueling service project on January 12, 2022. Cooperation between members of the logistics services supply chain (acronyms are LSSC) reduces, to some extent, the transaction costs of horizontal competition between chain members. Seo et al. [6] investigated information sharing, knowledge creation, goal similarity, decision coordination, and joint supply chain performance measurement as factors supporting supply chain collaboration between ports and port users, such as shipping lines, inland transportation companies, freight forwarders, ship management companies, and third-party logistics companies. Seo et al. [7] collected 178 maritime transports from Korean logistics companies, analyzed the questionnaires, and proved that there is a positive impact on supply chain collaboration for maritime logistics services, and inter-firm cooperation in turn helps to improve port performance.

However, we observe that most LSIs and LSPs do not have permanent and fixed cooperation with each other, which creates a decentralized structure. If there were no strategic interaction, then the manufacturer would prefer vertical integration to decentralization [8]. Is it possible that decisions about how to integrate logistics services supply chain (LSSC) members are influenced by the service factors and competitive intensity? LSSCs differ significantly from typical supply chains in that most clients choose service aspects such as dependability, on-time delivery, and safety over price. Yap [9] mentions billing accuracy, responsiveness to customer requirements, ensuring cargo security, reliability of dispatch, sailing time to destination ports, connectivity offered, and frequency of sailing. Operators will be able to give their clients better service because of service provider differentiation. It is also important for LSIs to gain a competitive advantage by partnering with LSPs that offer high-quality services [10]. Customers with different product characteristics have different priorities in terms of service quality [11,12]. Depending on the priority of service quality, the types of customers can be divided into those sensitive to service efficiency, those sensitive to cargo security, those sensitive to billing accuracy, etc. This paper only targets a single type of customer for the study. This study focuses on customers sensitive to service efficiency, there is more literature on the impact of service efficiency on single supply chain, and the main related literature is described in detail in the second part of the literature review of this paper. In the case of service efficiency competition between chains, there is less literature on the influence of service efficiency on supply chain integration decision. Seo et al. [8] and Xia et al. [13] studied structural choice under the influence of service level and price based on competitive supply chain markets. The focus is on the effect of service competition and price elasticity on structural choice. In order to better meet the needs of service-efficiency-sensitive customers, service providers invest in service efficiency improvement. Therefore, based on the previous research and compared to previous research, we focus on the service efficiency and competition intensity of logistics companies and ask two key research questions: (1) What strategies should supply chain managers adopt when the service efficiency improvement coefficient is large and the competition intensity between chains is different? What are the effects of different integration decisions on the profits of self-chain and competitive chain? (2) What strategies should supply chain managers adopt when the service efficiency improvement coefficient is small and the competition intensity between chains is different? What are the effects of different integration decisions on the profits of self-chain and competitive chain? Based on the aforementioned information and important considerations, we examine the connection between LSSC's integration choices and stakeholders' profitability and determine the best

equilibrium method. The important contribution of this paper is to address the issue of how to make integration decisions for supply chains when service providers adopt service efficiency improvement and supply chains are competitive in the presence of customer demand for service efficiency sensitivity. Compared to previous research, this paper analyzes the impact of service competition intensity and service efficiency factor on the integration decision of the supply chain when both factors are present. The paper provides an important theoretical basis for further exploring the decision choice of longitudinal integration of competitive logistics service supply chains.

The remainder of the study is organized as follows: Section 2 provides a brief review of the relevant literature. Section 3 introduces the models, notation, and assumptions and describes some of the associated issues. We analyze the choice contexts for three distinct integration models in Section 4. In Section 5, By comparing the decision situations of the different integration models, we derive the equilibrium strategy, optimal strategy, and LSSC profitability. We provide three practical implications in Section 6 to wrap up the discussion. For clarity, we have provided all proofs in the Appendix A.

## 2. Review of the Literature

In this chapter, we examine the pertinent literature on supply chain rivalry and integration decisions from the perspectives of sustainability, conventional supply chains (product supply chains comprising producers and retailers), and supply chains for logistics services.

### 2.1. Research Related to Sustainability

There are many studies on supply chains in the context of sustainable development. The studies relevant to this paper focus on both economic sustainability and ecological sustainability. Lazar et al. [14] emphasizes primary and secondary links of investigated studies with 17 United Nations sustainable development goals. The study defines focus with integrating environmental, social, and economic sustainability for logistics- and supply-chain-related studies. The bibliometric analysis also examined keyword relations. One of the main contributions is that economic sustainability was identified as the most represented one-dimensional sustainability focus. Guimarães et al. [15] suggest that the main drivers involved in sustainable supply chain management of the industry are social responsibility, economic performance, regulations (environmental, regional, international), and the adoption of an innovative business model. Zhang, X. et al. [16] explore the effect of supply chain integration on the operational performance of an internet-based online business based in China. The results indicate that integrating different aspects of the supply chain positively impacts the operating performance, improving the financial performance of the companies involved in the integration process. The effective integration of enterprise resources can improve the profitability of the supply chain and also contribute to the sustainable development of the supply chain.

A supply chain is considered sustainable if it fully integrates ecologically responsible business practices into an effective and competitive model [17,18]; Ghosh et al. [19] mention that companies involved in effective supply chain management are lean because they use resources sparingly and produce less waste. Supply chain management can change the innovation strategy of the supply chain in corporate business [20]. Sustainability activities need to extend from raw material procurement to supply chain operations [21]. Peng, Y. et al. [22] note that the cooperation strategy in a green supply chain is influenced by the relationship between green marketing and customer satisfaction. Li et al. [23] examine internal and external green supply chain management activity on automobile performance: environmental performance, operational performance, positive economic performance, and negative economic performance. A closed-loop supply chain is the process of adding a reverse supply chain to the traditional forward supply chain. It has a positive effect on reducing environmental pollution. Song et al. [24] studied the issue of recycling channel selection for waste electrical and electronic equipment. The maritime

supply chain is a colossal ecosystem and the interface of the intercontinental trade market. Within this ecosystem, freight transportation is considered a fundamental component of all supply chain systems. As a matter of its demanding multimodal and intermodal character, freight transportation is a highly competitive market where actors involved, demand reliable and high-quality services at competitive prices [25]. Therefore, in the competitive supply chain environment, the effective integration of enterprise resources is conducive to environmentally and ecologically sustainable development.

### 2.2. Research Related to Traditional Supply Chain Competition

The main research questions regarding supply chain rivalry fall into two categories, the first of which is the coordination of competition among supply chain members in a single supply chain. Research on supply chain competition is prevalent and takes varied forms.

To analyze the effects of manufacturer channel encroachment on logistics integration and the overall supply chain, He et al. [26] investigated the logistics integration of an e-commerce platform service supply chain consisting of a manufacturer, an e-commerce platform, and a third party LSP. Wang [27] explored a supply chain consisting of a supplier who invests in innovation and a manufacturer who sells the product to the user, wherein the innovation increases the value of the product to the user. The degree to which suppliers spend on innovation is correlated with negotiating power according to an examination of bargaining power. Thomas et al. [28] examined the impact of supplier innovation on supply chain integration and sustainability performance and whether supply chain integration mediates between supply chain innovation and sustainability performance. They constructed a structural equation model with trust as a determinant of supplier innovation. The findings confirmed that supplier innovation competence and trust are the major determinants of supply chain integration and sustainability performance. However, when supply chain integration is included in the model, this influence is lost. Shen et al. [29] constructed a cross-border supply chain with LSPs and investigated the decision-making of product price and logistics service level under two models of service leadership, thereby highlighting the significance of LSPs' involvement in the cross-border supply chain's decision-making. The shipping alliance impact and vertical contract competition selection problem were studied by Liu [30] and Wang [5], respectively. These works of literature are relatively rich and mature in terms of theoretical systems and modeling methods, which provide the theoretical basis and models for this study.

We concentrated on the coordination between two chain stakeholders when there is the competition between two supply chains, which is related to another research question in supply chain competition. Wang et al. [31] considered chain-to-chain competition between two supply chains and studied competitive and sustainable supply chain network design. In the competitive phase, the equilibrium of the retail price and carbon emissions was investigated. A Benders decomposition algorithm was proposed to address the computational complexity of the large-scale problem. A modeling approach to analyzing inter-supply-chain rivalry in global supply networks in the presence of trade policies was provided by Feng et al. [32]. In the context of globalization, cross-border transactions are growing. To explain inter-chain competition under the Cournot–Nash model and demonstrate how trade policies such as tariffs, quotas, and subsidies impact the equilibrium of global supply chains, they developed an optimization model for each PM chain and used variational inequality theory to provide equilibrium conditions. Zhang et al. [33] considered a two-channel supply chain network consisting of numerous rival manufacturers, rival retailers, and rival demand marketplaces. Each manufacturer produces and sells their products through direct e-commerce channels and traditional physical channels. Although the store offers only offline services to customers, the manufacturer offers services to customers through both channels. The study found that the level of service in each channel was positively correlated with its transaction volume. In active and inactive e-commerce trans-

actions, the same equilibrium decision (service level, price, or profit) may exhibit opposing trends in terms of cross-channel price coefficients.

The above literature on supply chain competition in the context of integration decision research focuses on the competition between product supply chains and dual channel competition between product and service supply chains. In the logistics sector, service efficiency competition between chain-to-chain LSIs now outweighs price competition as the primary type of horizontal competition [5]. We believe that in the logistics industry, there is vertical competition among peer companies and horizontal competition between the interests of supply chain members and that the LSSC is a service chain wherein customers focus on service. Therefore, we create the Stackelberg model to characterize the essential traits of supply chains in logistics services, including service sensitive customer demand, upstream service providers' investment in service efficiency, and supply chain service efficiency competition.

### 2.3. Research Related to Supply Chain in Logistics Services

With the depth and sophistication of supply-chain-related research, many academics have begun investigating decision-making in specialized industries' use of LSSC. The research on LSSC is extensive and focuses on information management, operational decision-making, benefits coordination, and capital management of LSSC among other aspects. In the context of demand renewal, Liu et al. [34] examined the influence of loss-aversion preferences on service capacity sourcing choices in an LSSC composed of an LSI and a functional LSP. Lin et al. [35] proposed a structural mapping of the platform service supply chain that differs from the traditional service supply chain structure by constructing a sustainable management framework for the platform service supply chain based on an analysis of Chinese logistics industry data. An LSSC with one LSI and two rival functional LSPs engaging in the allocation was the subject of an investigation by Liu et al. [36]. Wei and Chang [37] used the Hotelling model, wherein a two-oligarch game was developed to test the effectiveness of price matching and logistics service enhancement in improving profitability, driven by an increasingly competitive market in which traditional and online retailers sell the same products to consumers. By analyzing the macro and micro environments and choosing appropriate research tools based on the structural–logical filtering of external and internal factors, Pogodina et al. [38] developed a methodology for monitoring the process of competitive strategy formation in transport logistics companies. Wang et al. [39] considered a market consisting of offline businesses, e-commerce platforms, and third party LSPs. Cao et al. [40] investigated the selection of service providers and order allocation for different procedures under the mass customization logistics service model, and formulated a nonlinear, mixed integer, multi-objective optimization model. Using quantitative approaches, the ideal CODP site, order allocation strategy, and supplier selection strategy were identified concurrently. The multi-objective programming model is transformed into a single-objective model through different methods, and an improved genetic algorithm based on multilayer coding techniques is designed to solve the model. Numerical tests demonstrate the effectiveness of the approach used in this research in solving the problem. The above research on LSSC enriches the research content of LSSC and provides theoretical guidance for the research of this study. In the context of integration decision making, this study focuses on the LSSC content.

The LSSC is a vehicle for vertical integration, and academics have studied the decision of vertical integration in LSSCs. The effects of vertical integration between terminal operators and shipping corporations on port capacity, port fees, market output, and consumer surplus were examined by Zhu et al. [41] in the shipping sector. Additionally, they designed a model to study the impact of vertical integration, with a focus on shipping companies' investment in port capacity. Liu et al. [42] emphasized the interplay between smart manufacturers and their smart logistics transformation. A Stackelberg game model was created based on the interactions and constraints between manufacturers and LSPs during the smart logistics transformation process so that LSPs can lower their service costs

by creating appropriate contracts that result in complete supply chain coordination. Paridaens and Notteboom [43] analyzed Maersk Line, CMA CGM, and MSC in the context of logistics integration and conducted an empirical study; the study confirms intra-carrier and inter-carrier variability in the time path of logistics integration, the spatial coverage when implementing integration initiatives, and the methods of implementation. An extensive overview of the integration of shipping logistics is provided. Jafari et al. [44] studied the moderating impacts of logistics integration and demand uncertainty on retail business performance using a sample of 261 stores in Sweden. The presence of high or low-demand uncertainty may not always be conducive to achieving logistical flexibility and thus better performance when applying delays. Additionally, merchants should not always anticipate better performance gains from the flexibility advantages of delay if they prioritize logistics integration. Prassida and Hsu [45] examined the mechanisms by which channel integration and logistics services affect customer satisfaction and their willingness to repurchase. The study discovered that different blended experiences had varying contributions to improving transaction-specific satisfaction from perceived channel integration quality and logistical service quality.

The distinction is that in this study—by building two parallel, competing LSSCs—this paper analyzes the impact of service competition intensity and service efficiency factor on the integration decision of the supply chain when both factors are present. Additionally, we discover that when two LSSCs make opposing choices, under certain circumstances, profits are gained by both. More interestingly, we also find that under certain conditions, two LSSCs may exist in a state of prisoner's dilemma.

## 3. Problem Description and Underlying Assumptions

In order to better meet the needs of service efficiency-sensitive customers, logistics service providers are investing in logistics service efficiency. While logistics service integrators operate in a fully competitive market and are subject to pressure from related businesses in terms of horizontal competition, should they adopt a vertical integration strategy for LSPs to make the supply chain more profitable? This study constructs two parallel competing two-level LSSCs wherein each chain consists of an LSI and LSP. The two LSSCs are oriented toward the same logistics service market. For instance, for the same type of market, the supply chains of the French shipping business Delta and the Danish shipping company Maersk are rivals. In Maersk's supply chain, Maersk Line serves as the LSI and Qingdao Port International Group, which offers port services in China, serves as the LSP. Similarly, in Delta Air Lines' supply chain, Delta Air Lines serves as the LSI and Shanghai International Port Group, which offers port services in China, is the LSP. Each secondary LSSC can choose either non-integration or integration, which corresponds to two LSSC structures: decentralized (D) and centralized (C). The LSP and LSI within the LSSC will play the Stackelberg game maximization in accordance with their respective profit maximization decisions when the LSSC takes the non-integration decision. When the LSSC chooses to integrate, a corresponding centralized structure emerges wherein the LSSC will provide a set of logistics services to the outside world with the LSI as the core. Therefore, theoretically, there are four possible combinations of LSSC structures based on the different integration strategies of the two LSSCs (where *i* is 1,2): the CC model (both LSSCs choose to integrate), CD model ($LSSC_1$ chooses to integrate, whereas $LSSC_2$ chooses to not integrate), the DC model ($LSSC_1$ chooses not to integrate, whereas $LSSC_2$ chooses to integrate), and the dual decentralized (DD) model (both LSSCs choose not to integrate) according to (Table 1).

**Table 1.** Four structural combinations of two secondary LSSCs.

|  | $LSSC_1$ Option Integration | $LSSC_1$ Selective Dispersion |
|---|---|---|
| $LSSC_2$ option integration | CC mode | DC mode |
| $LSSC_2$ chooses not to integrate | CD mode | DD mode |

It is presumptively true that the competition between two secondary LSSCs is Nash competition, which means that $LSSC_1$ and $LSSC_2$ have equal bargaining power or market position and would simultaneously implement their respective optimal strategies. Due to the symmetry of the integration model, the CD and DC models are symmetrical to each other in terms of competition structure; therefore, only three secondary LSSC competition models—the dual centralized (CC), hybrid (CD), and DD models—are considered in this work. As shown in Figure 1, the dashed boxes in the figure represent centralized structures that take integration decisions.

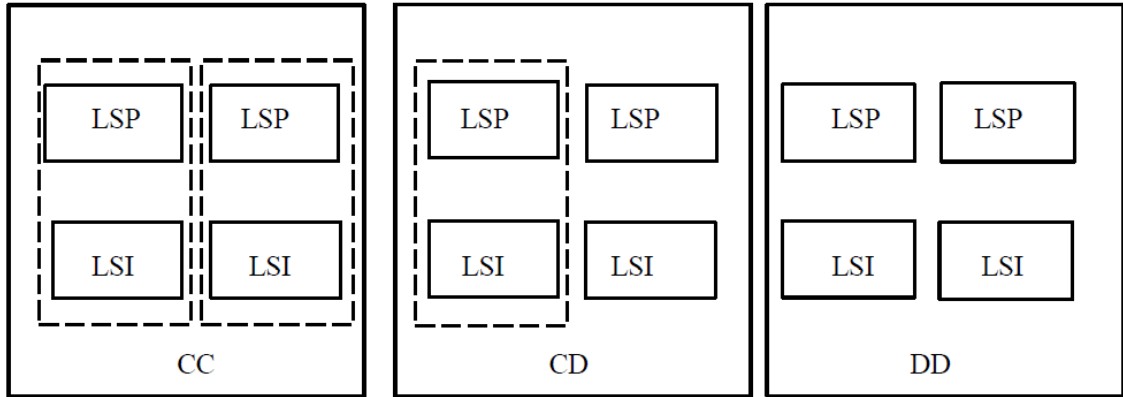

**Figure 1.** Secondary LSSC inter-competition model.

The LSPs and LSIs mutually decide on the level of investment in service efficiency and the price to be provided to clients when the supply chain members are centralized. When the supply chain members are decentralized, the supply chain members play a Stackelberg dynamic game wherein the sequence of decisions is as follows: due to market competition for service innovation, the LSP determines the degree of investment in service efficiency and the wholesale unit pricing of logistics services. Then, the LSI determines the price to be supplied to the customer to maximize its profit.

Following Wang and Liu's [5] model, $q_i = a - p_i + e_i - ue_{3-i}, i = 1, 2$ we set the demand function for logistics services $LSSC_i$ as

$$q_i = a - p_i + be_i - ube_{3-i}, i = 1, 2$$

Here, $q_i$ represents the market order volume of $LSSC_i$, potential demand for logistics service, $p_i$, and $e_i$ and $e_{3-i}$ represent the logistics service efficiency in different logistics service supply chains respectively, $be_i$ and $be_{3-i}$ the supply chain profit increase brought by the logistics service efficiency investment, $b$—that is, the coefficient of logistics service efficiency investment to cut unit cost, also known as unit improvement coefficient, which is $b = 1$ for the convenience of calculation in this paper. $u \in [0,1]$ is the service prices of logistics in various supply chains for logistics services, and represents the substitution coefficient between two LSSCs. The greater the value it takes, the fiercer the inter-chain competition.

In addition, this study assumes that the unit operating cost of LSI is $c_{LSI}$ and that the unit operating cost of LSP is $c_{LSP}$, and the information between LSP and LSI is symmetric and both are risk neutral, but the contract between the two LSSCs is unpredictable. This

study assumes that the service efficiency coefficients of both chains are equal to $k, k > 0$, The corresponding service efficiency investment cost is $ke_i^2$. The smaller $k$ means that the service cost to be invested in the same service efficiency is smaller; the larger $k$ means that the service cost to be invested in the same service efficiency is larger.

This study does not take into account for the particular kind of profit sharing used by the integrator; instead, it considers the two parties involved in the integration as a whole.

The symbols covered in this study and their meanings are shown in Table 2.

**Table 2.** Description of variables and symbols.

| Symbols | Meaning |
|---|---|
| $LSSC$ | Logistics services supply chain |
| $LSI_i$ | Logistics services integrator $i$ |
| $LSP_i$ | Logistics service provider $i$ |
| $c_{LSI}$ | Unit operating costs for integrators |
| $c_{LSP}$ | Unit operating costs for service providers |
| $\Pi_{LSP_i}$ | Profit for Service provider $i$ |
| $\Pi_{LSI_i}$ | Profit for integrator $i$ |
| $\Pi_{LSSC_i}$ | Total profit for chain $i$ |
| $q_i$ | Number of orders for logistics services in chain $i$ |
| $p_i$ | Chain $i$'s logistics services market unit price |
| $e_i$ | Level of service investment by service provider $i$ |
| $h_i$ | Wholesale unit prices for logistics services from service providers $i$ |
| $a$ | Potential basic needs in the logistics market |
| $u$ | The intensity of price competition among different integrators |
| $k$ | Service efficiency investment factor for service providers |
| $b$ | Coefficient of unit cost reduction for logistics services investment, also known as a unit service efficiency improvement factor |

Finally, let $\Pi_{z_i}^y, i = 1, 2$ represent the optimal profit of member $z$ of chain $i$ under model $y$. $z \in \{LSI_i, LSP_i\}$ denotes $LSI_i$ and $LSP_i$, respectively. $y \in \{CC, CD, DD\}$ denotes the three LSSC models under inter-chain Nash competition. These are the expressed equilibrium solutions: $e_i^y$ indicates the supply chain service providers' level of investment in service efficiency $y$; $h_i^y$ denotes the wholesale unit price of provider firms of supply chain $i$ under model $y$; $p_i^y$ indicates the market unit pricing of supply chain's under model $y$ as well as the quantity of service orders and represents the market unit price for logistics services according to model $y$; $q_i^y$ denotes the number of service orders for supply chain $i$ under model $y$.

## 4. Decision Making in Different Integration Decision Models

### 4.1. Dual Concentration Model (CC)

Both LSSCs select an integration plan with a centralized organization under the CC model. The specific game process is as follows: To increase total profit, both LSSCs choose a centralized structure in which $LSP_i$ and $LSI_i$ may jointly decide the level of logistics services to be offered externally, $e_i$, and the unit cost of logistics services, $p_i$. It is simple to determine that given a centralized structure, chain $i$'s objective profit function is

$$\Pi_{LSSC_i} = (p_i - c_{LSP} - c_{LSI})q_i - ke_i^2 \tag{1}$$

Substituting the demand function and solving optimally for $\Pi_{LSSC_i}$, the first-order partial derivatives $\Pi_{Pi}$ with respect to $p_i$ and $e_i$ are

$$\frac{\partial \Pi_{LSSC_i}}{\partial p_i} = a + c_{LSI} + c_{LSP} + e_i - ue_{3-i} - 2p_i \tag{2}$$

$$\frac{\partial \Pi_{LSSC_i}}{\partial e_i} = p_i - 2ke_i - c_{LSI} - c_{LSP} \tag{3}$$

Considering the Hesse matrix negative definite, it follows that $\frac{\partial^2 \Pi_{LSSC_i}}{\partial p_i^2} * \frac{\partial^2 \Pi_{LSSC_i}}{\partial e_i^2} - \frac{\partial^2 \Pi_{LSSC_i}}{\partial p_i e_i} * \frac{\partial^2 \Pi_{LSSC_i}}{\partial e_i p_i} > 0$, so it follows that $k > \frac{1}{4}$.

The equilibrium solution of the optimal chain *i* will be produced by the joint solution if Equations (2) and (3) are set to equal 0. By substituting the profit function into its target to determine its optimal profit, we may derive the following claim.

Proposition 1: Given the parameters a, $c_{LSI}$, $c_{LSP}$, $k$, and $u$, the optimal results of the CC competition model are as follows.

1. The equilibrium solution is

$$p_i^{CC} = \frac{2ka + (c_{LSI} + c_{LSP})(2k + u - 1)}{4k + u - 1} \tag{4}$$

$$e_i^{CC} = \frac{a - c_{LSI} - c_{LSP}}{4k + u - 1} \tag{5}$$

$$q_i^{CC} = \frac{2k(a - c_{LSI} - c_{LSP})}{4k + u - 1} \tag{6}$$

2. The profit of the two LSIs, i.e., the optimal profit of the two LSSCs, is

$$\Pi_{LSSC_i}^{CC} = \Pi_{LSI_i}^{CC} = \frac{k(4k - 1)(a - c_{LSI} - c_{LSP})^2}{(2k + u - 1)^2} \tag{7}$$

*4.2. Hybrid Model (CD)*

The mixed competition structures CD and DC are of essentially the same type under inter-chain Nash competition due to the symmetry of the structures, with one LSSC choosing a decentralized structure and the other a centralized structure. For generality, it is assumed that $LSSC_1$ adopts a centralized structure, i.e., together, $LSI_1$ and $LSP_1$ form an integrated system that serves the logistics service market by offering. $LSSC_2$ adopts a decentralized structure, i.e., $LSP_2$ sets the degree of logistics services given to its downstream customers, $LSI_2$ establishes the market pricing of logistics services for its clients, and the two companies work together to form $LSSC_2$. Horizontal competition occurs between $LSSC_1$ and $LSSC_2$, and Nash competition occurs between $LSSC_1$ and $LSSC_2$, thereby constituting a mixed competition model (CD).

The specific game's procedure is as follows: in the first stage, $LSP_2$ decides the level of service and at what wholesale unit price to deliver to $LSI_2$ downstream for traditional services; in the second stage, $LSI_2$ downstream determines the logistics service market's unit price $p_2$, whereas the integrator of $LSSC_1$ jointly determines its logistics service market's unit price $p_1$ and service level $e_1$ to finally maximize the profit of each member. This means that in the hybrid structure, it is possible to achieve the objective profit functions of $LSSC_1$ and $LSSC_2$ under the hybrid structure are as follows:

The objective profit function for $LSSC_1$ is

$$\Pi_{LSSC_1} = (p_1 - c_{LSP} - c_{LSI})q_1 - ke_i^2 \tag{8}$$

The objective profit function for $LSSC_2$ is

$$\Pi_{LSP_2} = (h_2 - c_{LSP})q_2 - ke_2^2 \tag{9}$$

$$\Pi_{LSI_2} = (p_2 - h_2 - c_{LSI})q_2 \tag{10}$$

The demand function is substituted, and the backward derivation is used to solve for the objective profit function of each of the two LSSCs. $\Pi_{C_2}$ is first optimized so that the first-order derivative of with respect to $\Pi_{LSI2}p_2$ is 0 to obtain the reaction functions of $p_2$ with respect to $h_2$ and $e_2$

$$p_2 = \frac{1}{2}(a + e_2 + h_2 + c_{LSI} - ue_1) \tag{11}$$

Furthermore, we optimize $\Pi_{LSSC_1}$ by finding the first-order partial derivatives $\Pi_{LSSC_1}$ with respect to $p_1$ and $e_1$, thereby making them 0 to obtain

$$p_1 = \frac{2k(a - ue_2) + (2k - 1)(c_{LSI} + c_{LSP})}{4k - 1} \tag{12}$$

$$e_1 = \frac{a - ue_2 - c_{LSI} - c_{LSP}}{4k - 1} \tag{13}$$

Substituting the reaction function $p_2$ into the upstream $\Pi_{LSP_2}$ and finding the first-order partial derivatives of $\Pi_{LSP_2}$ concerning $h_2$ and $e_2$ to 0 gives

$$h_2 = \frac{4k(a - ue_1) - 4kc_{LSI} + (4k - 1)c_{LSP}}{8k - 1} \tag{14}$$

$$e_2 = \frac{a - ue_1 - c_{LSI} - c_{LSP}}{8k - 1} \tag{15}$$

From the negative definite of the Hesse matrix, we obtain $k > \frac{1}{4}$. The final joint Equations (11)–(15) lead to the equilibrium solutions of the optimal $LSSC_1$ and $LSSC_2$, which are substituted into their objective profit functions to find their optimal profits, thereby yielding the following proposition.

Proposition 2: Given parameters $a$, $c_{LSP}$, $c_{LSI}$, $k$, and $u$, the optimal results of the CD competition model are as follows.

The equilibrium solution is

$$p_1^{CD} = \frac{2k(8k - 1 - u)a + (1 + 16k^2 - 2k(5 - u) - u^2)(c_{LSI} + c_{LSP})}{1 + 32k^2 - 12k - u^2} \tag{16}$$

$$e_1^{CD} = \frac{(8k - 1 - u)(a - c_{LSI} - c_{LSP})}{1 + 32k^2 - 12k - u^2} \tag{17}$$

$$q_1^{CD} = \frac{(8k - 1 - u)(a - c_{LSI} - c_{LSP})}{1 + 32k^2 - 12k - u^2} \tag{18}$$

$$p_2^{CD} = \frac{6k(4k - 1 - u)a + (1 + 8k^2 - 6k(1 - u) - u^2)(c_{LSI} + c_{LSP})}{1 + 32k^2 - 12k - u^2} \tag{19}$$

$$h_2^{CD} = \frac{4k(4k - 1 - u)(a - c_{LSI}) + (1 + 16k^2 - 4k(2 - u) - u^2)c_{LSP}}{1 + 32k^2 - 12k - u^2} \tag{20}$$

$$e_2^{CD} = \frac{4k(4k - 1 - u)(a - c_{LSI} - c_{LSP})}{1 + 32k^2 - 12k - u^2} \tag{21}$$

$$q_2^{CD} = \frac{2k(4k - 1 - u)(a - c_{LSI} - c_{LSP})}{1 + 32k^2 - 12k - u^2} \tag{22}$$

The optimal profit for each of the two LSSC members is

$$\Pi_{LSI1}^{CD} = \frac{k(4k-1)(8k-1-u)^2(a-c_{LSI}-c_{LSP})^2}{(1+32k^2-12k-u^2)^2} \tag{23}$$

$$\Pi_{LSI2}^{CD} = \frac{k(8k-1)(4k-1-u)^2(a-c_{LSI}-c_{LSP})^2}{(1+32k^2-12k-u^2)^2} \tag{24}$$

$$\Pi_{LSP2}^{CD} = \frac{4k^2(2k-1-u)^2(a-c_{LSI}-c_{LSP})^2}{(1+32k^2-12k-u^2)^2} \tag{25}$$

The total optimal profit for each of the two LSSCs is

$$\Pi_{LSSC1}^{CD} = \frac{k(4k-1)(8k-1-u)^2(a-c_{LSI}-c_{LSP})^2}{(1+32k^2-12k-u^2)^2} \tag{26}$$

$$\Pi_{LSSC2}^{CD} = \frac{k(12k-1)(4k-1-u)^2(a-c_{LSI}-c_{LSP})^2}{(1+32k^2-12k-u^2)^2} \tag{27}$$

The equilibrium solutions and optimal profits of the hybrid structures DC and CD are numerically symmetric to each other and will not be repeated here.

### 4.3. Double Dispersion Model (DD)

There is horizontal Nash competition between the two chains when both LSSCs decide against integrating, i.e., when $LSP_1$ and $LSI_1$ form $LSSC_1$ and $LSP_2$ and $LSI_2$ form $LSSC_2$, they create a double decentralized competition model (DD). The particular game mechanics are as follows: To optimize its profit, $LSI_i$ faces the market after $LSP_i$ determines the best wholesale unit price for logistics services $h_i$ and service quality. After considering the market, $LSI_i$ chooses its unit costs for logistics services.

This enables the following objective profit function to be derived for chain $i$ under a double decentralized structure.

$$\Pi_{LSP_i} = (h_i - c_{LSP})q_i - ke_i^2 \tag{28}$$

$$\Pi_{LSI_i} = (p_i - h_i - c_{LSI})q_i \tag{29}$$

Substitute the demand function and solve the objective profit function for chain $i$ using the inverse derivative method. First, optimize $\Pi_{LSI_i}$ so that the first-order derivative $\Pi_{LSI_i}$ with respect to $p_i$ is 0 to obtain the reaction function of $p_i$ with respect to $h_i$ and $e_i$

$$p_i = \frac{1}{2}(a + e_i + h_i + c_{LSI} - ue_{3-i}) \tag{30}$$

Substituting the reaction function $\Pi_{LSP_i}$ upstream and finding the first-order partial derivatives $\Pi_{LSP_i}$ with respect $h_i$ and $e_i$ to 0 gives

$$h_i = \frac{4k(a - ue_{3-i}) - 4kc_{LSI} + (4k-1)c_{LSP}}{8k-1} \tag{31}$$

$$e_i = \frac{a - c_{LSI} - c_{LSP} - ue_{3-i}}{8k-1} \tag{32}$$

Using the Hesse matrix negative definite, i.e., $\frac{\partial^2\Pi_{LSPi}}{\partial h_i^2} * \frac{\partial^2\Pi_{LSPi}}{\partial e_i^2} - \frac{\partial^2\Pi_{LSPi}}{\partial h_i e_i} * \frac{\partial^2\Pi_{LSPi}}{\partial e_i h_i} > 0$, we obtain $k > \frac{1}{8}$.

Ultimately, using Equations (30)–(32), the equilibrium solution of the optimal chain $i$ can be obtained by substituting its objective profit function and finding its optimal profit, the following proposition can be obtained.

Proposition 3: Given parameters $a$, $c_P$, $c_C$, $k$, and $u$, the optimal results of the DD competition model are as follows.

The equilibrium solution is

$$p_i^{DD} = \frac{6ka + (2k + u - 1)(c_{LSI} + c_{LSP})}{8k + u - 1} \tag{33}$$

$$h_i^{DD} = \frac{4k(a - c_{LSI}) + (2k + u - 1)c_{LSP}}{8k + u - 1} \tag{34}$$

$$e_i^{DD} = \frac{a - c_{LSI} - c_{LSP}}{8k + u - 1} \tag{35}$$

$$q_i^{DD} = \frac{k(a - c_{LSI} - c_{LSP})}{8k + u - 1} \tag{36}$$

The optimal profits of LSPs and LSIs on the two LSSCs, respectively, are

$$\Pi_{LSP_i}^{DD} = \frac{k(8k - 1)(a - c_{LSI} - c_{LSP})^2}{(8k + u - 1)^2} \tag{37}$$

$$\Pi_{LSI_i}^{DD} = \frac{4k^2(a - c_{LSI} - c_{LSP})^2}{(8k + u - 1)^2} \tag{38}$$

The total optimal profit for each of the two LSSCs is

$$\Pi_{LSSC_i}^{DD} = \frac{k(12k - 1)(a - c_{LSI} - c_{LSP})^2}{(8k + u - 1)^2} \tag{39}$$

## 5. Comparative Profit Analysis of Different Integration Decision Models

*5.1. Comparing the Profitability of a Chain When Rival Chains Choose Not to Integrate (D)*

This section uses the supply chains of Maersk Line and Delta Shipping as examples with practical understanding. The supply chain of Maersk Line is $LSSC_1$, and the supply chain of Delta Shipping is $LSSC_2$. The Maersk Shipping Company is referred to as $LSI_1$, and the Qingdao Port International Group is referred to as LSP1 in $LSSC_1$; in $LSSC_2$, $LSI_2$ refers to the Duffy Shipping Company and $LSP_2$ refers to the Shanghai International Group. Given that the Maersk chain has both integration and non-integration options and that the Duffy chain is the rival chain, the two LSSCs will progressively construct a CD structure and a DD structure when the Duffy chain decides not to integrate (D). At this point, the relationship between the magnitude of the overall profit of the Maersk chain under the DD and CD structures is compared, and the symmetry of the competitive model can be seen in values $\Pi_{T1}^{CC} = \Pi_{T2}^{CC}$, $\Pi_{T1}^{DD} = \Pi_{T2}^{DD}$, $\Pi_{T1}^{CD} = \Pi_{T2}^{DC}$, and $\Pi_{T1}^{DC} = \Pi_{T2}^{CD}$. The following citation can be obtained.

**Lemma 1.** *From* $\Pi_{LSSC_1}^{CD} - \Pi_{LSSC_1}^{DD} = 0$, *we have* $u = u_1(k)$, *and then we have the following*:

(i)    *if* $\frac{1}{4} < k < 0.3259$ *and* $0 < u < u_1(k)$, *then* $\Pi_{LSSC_1}^{CD} > \Pi_{LSSC_1}^{DD}$;

(ii)   *if* $\frac{1}{4} < k < 0.3259$ *and* $u_1(k) < u < 1$, *then* $\Pi_{LSSC_1}^{CD} < \Pi_{LSSC_1}^{DD}$; *and*

(iii)  *if* $k \geq 0.3259$ *and* $0 < u < 1$, *then* $\Pi_{LSSC_1}^{CD} > \Pi_{LSSC_1}^{DD}$.

See the Appendix A for the proof.

The results are shown in Figure 2. In region A1, $\Pi_{LSSC_1}^{CD} > \Pi_{LSSC_1}^{DD}$, and in region A2, $\Pi_{LSSC1}^{CD} < \Pi_{LSSC1}^{DD}$.

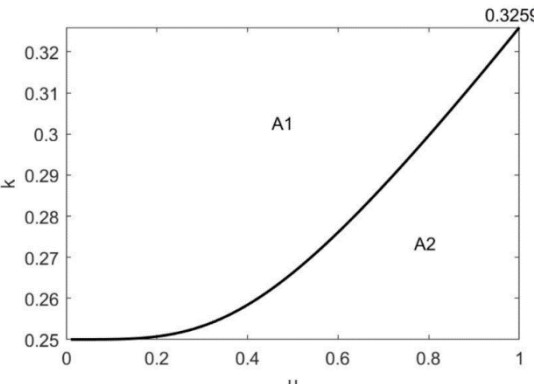

**Figure 2.** Relationship diagram for $\Pi_{LSSC_1}^{CD}, \Pi_{LSSC_1}^{DD}$.

From Lemma 1, it can be deduced that the Maersk Shipping Company and Qingdao Port International Group make more profit by taking the integration decision instead of the non-integration decision; the Daffodil Shipping Company and Shanghai International Group take a decentralized decision without integration when the service efficiency investment coefficient is smaller and the intensity of price competition between the chains is also small. Maersk Line and Qingdao Port International Group can make the Maersk chain more profitable by taking the non-integration decision than by taking the integration decision when the service efficiency investment coefficient is large, regardless of the price competition between the two chains.

The following theorem follows from Lemma 1.

**Theorem 1.** *For one's own LSSC, a competing LSSC's decision to integrate or not is based on the degree of competition between the two chains and the range of service efficiency investment cost considerations. Both requirements are met by*

(i)    *When $0.25 < k < 0.3259$ and $0 < u < u_1(k)$, its LSSC chooses integration;*
(ii)   *When $0.25 < k < 0.3259$ and $u_1(k) < u < 1$, its LSSC chooses not to integrate; and*
(iii)  *When $k \geq 0.3259$, $0 < u < 1$ holds constant and its LSSC chooses integration.*

Theorem 1 demonstrates that the LSI in a chain will decide whether to integrate with the upstream LSP depending on its service efficiency investment cost factor $k$ and service efficiency competition intensity $u$ when the competitor chain decides not to integrate. Figure 2 illustrates how area A1′s chain which decides to integrate can increase profits because at this time it has a smaller service efficiency investment cost factor but not as high service efficiency competition intensity. For the LSI in a chain, even if the rival chain opts for a decentralized structure, their chain tends to integrate its own upstream and downstream members to win market competition and increase overall profit; however, in region A2, their chain will earn higher profit if they choose not to integrate. Due to the severe competition from downstream LSIs, when the rival chain chooses a decentralized structure, its chain also opts to keep its LSPs and LSIs separate. In this manner, the upstream LSPs are spared from the fierce competition from downstream LSIs. When the service efficiency investment coefficient is large $k \geq 0.3536$, the LSI's decision to integrate with the upstream LSP mostly depends on the size of the service efficiency investment coefficient rather than the level of service efficiency competition. In other words, when the service efficiency investment coefficient is high, regardless of how fierce the service rivalry is, the rival chain opts not to integrate, whereas its chain will typically integrate its upstream and downstream members to increase profitability.

**Lemma 2.** *From $\Pi_{LSSC2}^{CD} - \Pi_{LSSC2}^{DD} = 0$, we have $u = u_2(k)$, and then we have the following:*
(i)    *if $0.25 < k < 0.4375$ and $0 < u < u_2(k)$, then $\Pi_{LSSC2}^{CD} < \Pi_{LSSC2}^{DD}$;*
(ii)   *if $0.25 < k < 0.4375$ and $u_2(k) < u < 1$, then $\Pi_{LSSC2}^{CD} > \Pi_{LSSC2}^{DD}$; and*

(iii)   *if $k \geq 0.4375$ and $0 < u < 1$, then $\Pi_{LSSC2}^{CD} < \Pi_{LSSC2}^{DD}$.*

The proof is similar to that of Lemma 1.

From Lemma 2, it follows that regardless of the level of price competition between the two chains, the Maersk Shipping Company and Qingdao Port International Group can increase the profitability of the Duffy chain more profitable by making a decentralized decision to integrate when the Duffy Shipping Company and Shanghai International Group decide against it. The Maersk Line and Qingdao Port International Group can increase the profits of the Duffy chain by adopting the integration option when the service efficiency investment factor is low and the intensity of inter-chain price rivalry fulfills specific requirements. The non-integration decision taken by the Maersk Line and Qingdao Port International Group results in higher profitability for the Duffy chain than the integration decision when the service efficiency investment factor is low and the level of inter-chain price rivalry fulfills certain parameters.

The results are shown in Figure 3. In region A3, $\Pi_{LSSC2}^{CD} < \Pi_{LSSC2}^{DD}$, and in region A4, $\Pi_{LSSC2}^{CD} > \Pi_{LSSC2}^{DD}$.

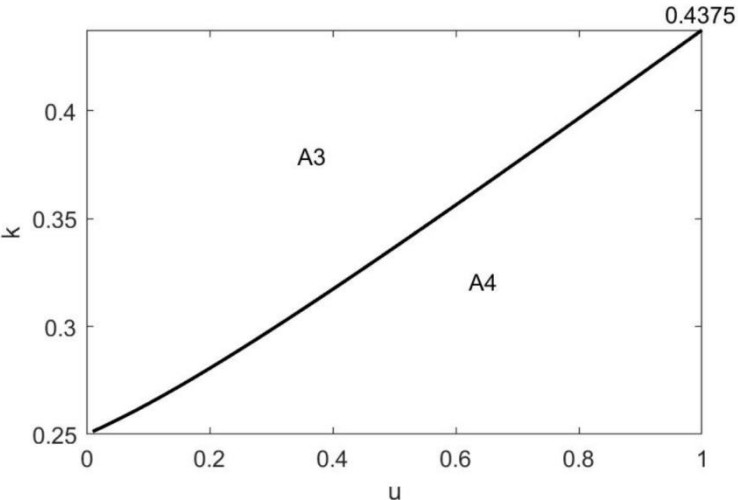

**Figure 3.** Relationship diagram for $\Pi_{LSSC2}^{CD}, \Pi_{LSSC2}^{DD}$.

The following theorem follows from Lemma 2.

**Theorem 2.** *It is known that the rival LSSC chooses not to integrate and that either option will have an equal influence on the overall profitability of the competing LSSC. The effect will vary depending on the level of competition between the two LSSCs and the range of service efficiency investment cost factors. Both parameters are satisfied as follows:*

(i)     *When $0.25 < k < 0.4375$ and $0 < u < u_2(k)$, the rival chain is more profitable if its chain chooses not to integrate;*
(ii)    *When $0.25 < k < 0.4375$ and $u_2(k) < u < 1$, the rival chain is more profitable if its chain chooses to integrate;*
(iii)   *When $k \geq 0.4375$ and $0 < u < 1$, the rival chain is more profitable if its chain chooses not to integrate.*

Theorem 2 demonstrates that when LSI price competition degree $u$ and LSP service efficiency investment coefficient $k$ change, one's own LSSC will have different integration strategies, and a change of strategy will also affect the overall profit of the rival LSSC in turn. This effect is assuming that the rival LSSC decides not to integrate. When the rival chain has been determined to be decentralized, as shown in Figure 3, the upper-left region A3 corresponds to a smaller service efficiency cost factor and smaller competitive

intensity. Compared to the structure wherein one's own chain does not integrate, the lower-right region A4 correlates to a lower service efficiency investment cost component and a higher level of competition, which will boost the overall profit of the rival chain. When the service efficiency investment factor is larger, $k \geq 0.4375$, the service efficiency investment coefficient and not the intensity of service efficiency competition determines whether LSIs in one's own chain choose to integrate with upstream LSPs. In other words, when the service efficiency investment coefficient is high, LSIs in their chain tend to integrate their own upstream and downstream members to increase the profitability of the rival chain regardless of the rival chain's decision not to integrate or the level of service efficiency competition.

Combining Figures 2 and 3, as shown in Figure 4, with $LSSC_2$ choosing decentralized decision-making, in region A5, $\Pi_{LSSC1}^{CD} > \Pi_{LSSC1}^{DD}$, $\Pi_{LSSC2}^{CD} < \Pi_{LSSC2}^{DD}$—i.e., $LSSC_1$ prefers centralized decision-making, which reduces $LSSC_2$'s profit; in region A6, $\Pi_{LSSC1}^{CD} > \Pi_{LSSC1}^{DD}$ $\Pi_{LSSC2}^{CD} > \Pi_{LSSC2}^{DD}$—i.e., $LSSC_1$ prefers centralized decision making, which increases $LSSC_2$'s profit; and in region A7, $\Pi_{LSSC1}^{CD} < \Pi_{LSSC1}^{DD}$, $\Pi_{LSSC2}^{CD} > \Pi_{LSSC2}^{DD}$—i.e., $LSSC_1$ prefers decentralized decision making, which increases $LSSC_2$'s profits.

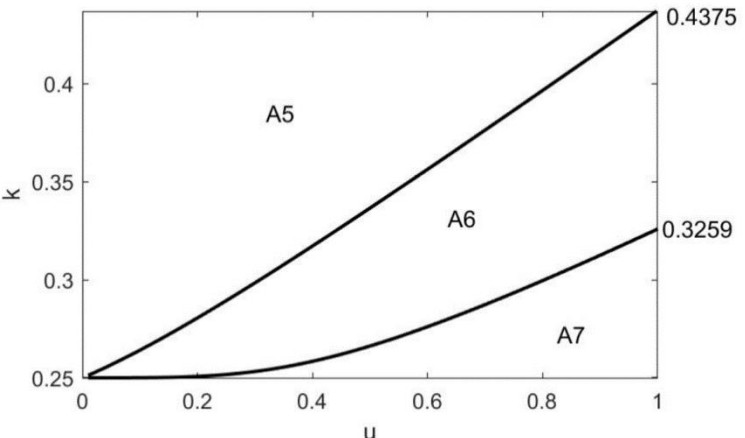

**Figure 4.** Diagram of the relationship between $\Pi_{LSSC1}^{CD}$, $\Pi_{LSSC1}^{DD}$ and $\Pi_{LSSC2}^{CD}$, $\Pi_{LSSC2}^{DD}$.

Assuming that $LSSC_2$ always maintains a decentralized structure, it can be seen from regions A6 and A7 in Figure 4 that when the efficiency investment cost coefficient gradually increases—i.e., from region A7 to region A6—it can be demonstrated that the optimal strategy of $LSSC_1$ changes from decentralized to centralized. Both chains' service efficiency investment costs must rise to a certain amount to maintain a particular level of service when the service efficiency investment cost coefficient $k$ rises. To obtain more integration benefits and allow the LSIs and LSPs to bear the burden of increasing service prices, $LSSC_1$ may combine its LSPs and LSIs. Within a specific $(k,u)$ range, i.e., region A6, such integration benefits will generate a certain degree of positive externality, thereby increasing the overall profits of counterparty $LSSC_2$ to some extent. $LSSC_1$ will continue to maintain its integration measures and strengthen integration benefits upstream and downstream as the efficiency investment cost factor rises, thereby moving from region A6 to A5. At this point, the competitive market advantage provided by the integration measures to its $LSSC_1$ will greatly outweigh its positive externality of spillover, thereby somewhat suppressing the competitive market position of the decentralized $LSSC_2$.

Consequently, the following inference can be drawn.

**Corollary 1**. *When (k,u) is in a particular range, two LSSCs with horizontal Nash competition occur at the same level, and when one LSSC chooses not to integrate, the other LSSC that chooses to integrate this choice not only enhances its overall profit but also contributes to the overall profit of its rival LSSC.*

**Corollary 2**. *Two LSSCs with horizontal Nash competition occur at the same level. When one LSSC chooses not to integrate, the other LSSC chooses not to integrate, thereby increasing the profitability of at most one of its chains or that of its rival chain.*

*5.2. Comparing the Profitability of a Chain When Rival Chains Choose to Integrate (C)*

When $LSSC_2$ chooses to integrate (C), the overall profit size relationship between $LSSC_1$ under the CC and DC structures is compared and the following corollary can be obtained.

**Lemma 3**. *follows from* $\Pi_{LSSC1}^{CC} - \Pi_{LSSC1}^{DC} = 0$, *which gives* $u = u_3(k)$, *which gives*

(i)    *If $0.25 < k < 0.4167$ and $0 < u < u_3(k)$, then $\Pi_{LSSC1}^{CC} > \Pi_{LSSC1}^{DC}$;*
(ii)   *If $0.25 < k < 0.4167$ and $u_3(k) < u < 1$, then $\Pi_{LSSC1}^{CC} < \Pi_{LSSC1}^{DC}$;*
(iii)  *If $k \geq 0.4167$ and $0 < u < 1$, then $\Pi_{LSSC1}^{CC} > \Pi_{LSSC1}^{DC}$.*

The proof is similar to that of Lemma 1.

From Lemma 3, it can be concluded that when the integration is carried out by the centralized decision of the Duffy Shipping Company and Shanghai International Group and when the service efficiency investment coefficient is small and the intensity of price competition between the chains is also small, the integration decision by the Maersk Shipping Company and Qingdao Port International Group can make the profits of the Maersk chain greater than those obtained by taking the non-integration decision; when the price competition between the chains is fierce and service efficiency investment factor is low, the Maersk Line and Qingdao Port International Group benefit more from choosing not to integrate than from integrating. However, when the service efficiency investment factor is high, both companies can benefit from integrating regardless of how fierce the price competition is. The results are shown in Figure 5, In region B1, $\Pi_{LSSC1}^{CC} > \Pi_{LSSC1}^{DC}$; in region B2, $\Pi_{LSSC1}^{CC} < \Pi_{LSSC1}^{DC}$.

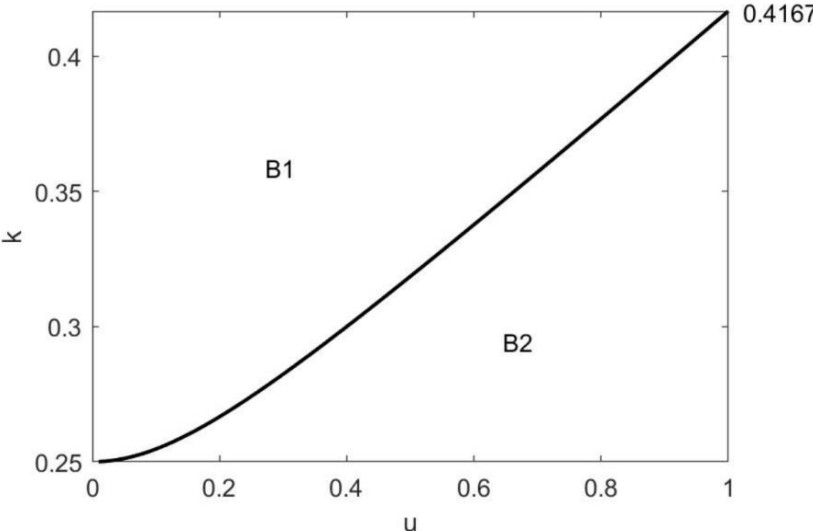

**Figure 5.** Relationship diagram for $\Pi_{LSSC1}^{CC}, \Pi_{LSSC1}^{DC}$.

The following theorem follows from Lemma 3.

**Theorem 3**. *The level of competition between the two LSSCs and the range of service efficiency investment cost factors elements determine whether a competing chain decides to integrate for its LSSC. Both parameters are satisfied by*

(i)     *When* $0.25 < k < 0.4167$ *and* $0 < u < u_3(k)$, *its LSSC selects integration;*
(ii)    *When* $0.25 < k < 0.4167$ *and* $u_3(k) < u < 1$, *LSSC chooses non-integration;*
(iii)   *When* $k \geq 0.4167$, $0 < u < 1$ *holds constant and its LSSC chooses integration.*

Theorem 3 demonstrates that the degree of competition u for the LSI in a rival chain and that service input cost factor *k* has an identical impact on a rival chain's decision to integrate or not with a downstream LSP.

Figure 5 demonstrates that in region B2, a lower service input cost component is equated with a higher level of competitiveness. Although the rival chain already has an integration strategy to obtain a certain degree of integration benefits, one's own chain has decided not to integrate at this time to compete in the market. This is likely because given the fierce market competition, adopting the same integration as the competitor chain may not result in more integration benefits; rather, it will subject the downstream LSPs to LSSC competition. The upstream LSPs would prefer to avoid the losses caused by severe LSSC rivalry; therefore, its chain would prefer not to integrate and to keep the original LSPs and LSIs separate. In region B1, a smaller service efficiency investment cost factor corresponds to a smaller degree of LSSC competition. The benefits of integration are not significantly harmed by competition, and the additional benefits of integration can offset the modest losses brought on by competition. When the service efficiency investment coefficient is large—$k \geq 0.4167$—for LSIs in their chain, the choice to integrate with upstream LSPs depends mainly on the service efficiency investment coefficient and not the intensity of service efficiency competition. To put it another way, the LSI in its chain should decide to integrate with upstream and downstream members when the service efficiency investment factor is high regardless of the level of service efficiency competition to increase its market competitiveness, increase profitability, and avoid falling behind rival chains in the logistics service market if the rival chains have already made this choice.

**Lemma 4**. *it is obtained from* $\Pi_{LSSC2}^{CC} - \Pi_{LSSC2}^{DC} = 0$ *to* $u = u_4(k) = k + \sqrt{1 - 12k + 33k^2}$, *which gives*

(i)     *If* $0.25 < k < 0.3125$ *and* $u_4(k) < u < 1$, *then* $\Pi_{LSSC_2}^{CC} > \Pi_{LSSC_2}^{DC}$;
(ii)    *If* $0.25 < k < 0.3125$ *and* $0 < u < u_4(k)$, *then* $\Pi_{LSSC_2}^{DC} > \Pi_{LSSC_2}^{CC}$;
(iii)   *If* $k > 0.3125$ *and* $0 < u < 1$, *then* $\Pi_{LSSC_2}^{DC} > \Pi_{LSSC_2}^{CC}$.

The proof is similar to that of Lemma 1.

From Lemma 4, it can be concluded that when the integration is carried out by the centralized decision of the Duffy Shipping Company and Shanghai International Group, When the service efficiency investment coefficient is small and the intensity of price competition between the chains is also small, the Maersk Shipping Company and Qingdao Port International Group can make the profits of Maersk chains greater by adopting the non-integration decision compared to the integration decision; when the service efficiency investment factor is small and the competition between the chains is strong, the Maersk Line and Qingdao Port International Group can make more profit by adopting the integration decision compared to the non-integration decision. When the service efficiency investment factor is large, the Maersk Line and Qingdao Port International Group can make more profit than the Duffy chain by adopting the non-integration decision regardless of the price competition between the two chains.

The results for region B3, $\Pi_{LSSC2}^{CC} < \Pi_{LSSC2}^{DC}$, and region B4, $\Pi_{LSSC2}^{CC} > \Pi_{LSSC2}^{DC}$, are shown in Figure 6. The following theorem follows from Lemma 4.

**Theorem 4**. *It is known that the rival express chain opts to integrate, and its decision to integrate or not will have an equal impact on the rival express chain's overall profitability. The impact will depend on the level of service efficiency competition between the two express chains as well as various service efficiency investment factors. Two parameters are satisfied by*

(i)     When $0.25 < k < 0.3125$ and $u_4(k) < u < 1$, the rival chain is more profitable if its chain chooses to integrate;

(ii)    When $0.25 < k < 0.3125$ and $0 < u < u_4(k)$, the rival chain is more profitable if its chain chooses not to integrate;

(iii)   When $k > 0.3125$ and $0 < u < 1$ hold, the rival chain is more profitable if its chain chooses not to integrate.

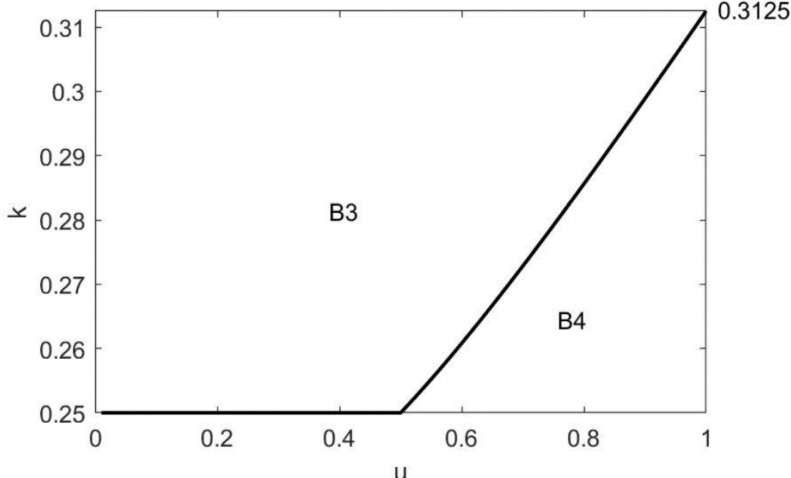

**Figure 6.** Relationship diagram for $\Pi_{LSSC_2}^{CC}, \Pi_{LSSC_2}^{DC}$.

As shown in Figure 6, region B3 corresponds to a smaller service efficiency investment coefficient and larger service efficiency competition intensity, and if the strategy of its chain is not integrated, it will increase the overall profit of the rival chain; region B4 corresponds to a smaller service efficiency investment factor and smaller service efficiency competition intensity. If the strategy of its chain is to integrate, compared to the structure of its chain without integration, it will improve the overall profit of the rival chain. When the service efficiency investment factor is larger—$k \geq 0.3125$—the service efficiency investment factor, not the size of the service efficiency competition intensity, determines whether the LSI in its chain decides to integrate with the upstream LSP. For example, when the service efficiency investment factor for the LSI in its chain is high, the competitor chain will typically opt not to integrate, and one's own chain will typically not integrate its own upstream and downstream members to make the rival chain more profitable.

At this point, Theorem 4 is combined with Theorem 3 to analyze the combined effects of service efficiency investment coefficient $k$ and service efficiency competition intensity $u$ on the integration decision of its chain and the change in the profitability of rival chains.

Combining Figures 5 and 6 into one—as shown in Figure 7—with Chain 2 choosing a centralized structure, in region B5, $\Pi_{LSSC1}^{CC} > \Pi_{LSSC1}^{DC} \Pi_{LSSC2}^{CC} < \Pi_{LSSC2}^{DC}$—i.e., Chain 1 prefers a centralized structure, which reduces Chain 2's profits; in region B6, $\Pi_{LSSC1}^{CC} < \Pi_{LSSC1}^{DC}, \Pi_{LSSC2}^{CC} < \Pi_{LSSC2}^{DC}$—i.e., Chain 1 prefers a decentralized structure, which increases Chain 2's profits; and in region B7, $\Pi_{LSSC1}^{CC} < \Pi_{LSSC1}^{DC}, \Pi_{LSSC2}^{CC} > \Pi_{LSSC2}^{DC}$—i.e., Chain 1 prefers a decentralized structure, which reduces Chain 2's profits.

As can be seen from Figure 7, when service efficiency investment coefficient $k$ is within a certain range, corresponding to the three regions in Figure 7 under the premise that Chain 2 always maintains a centralized structure, Chain 1 always maintains a decentralized structure as $k$ gradually increases—i.e., from region B7 to B6 within a specific range of $(k, u)$, Chain 1 can consider utilizing the positive externalities generated by Chain 2's integration measures. Chain 2 maintains a centralized structure to benefit from integration as k rises from region B6 to B5, whereas Chain 1 also chooses an integration strategy to compete. This is because both chains' service efficiency investment costs will rise as the service efficiency investment factor $k$ increases unless Chain 2 already has an integration plan to achieve

further integration benefits. Chain 1 will also adopt the integration strategy to further strengthen its competitive power, thereby causing the profits of its rival Chain 2 to decline because the positive externality of Chain 1 is insufficient to cover the service efficiency investment costs and shrinking profits of Chain 1 under the decentralized structure. The following inference can be drawn.

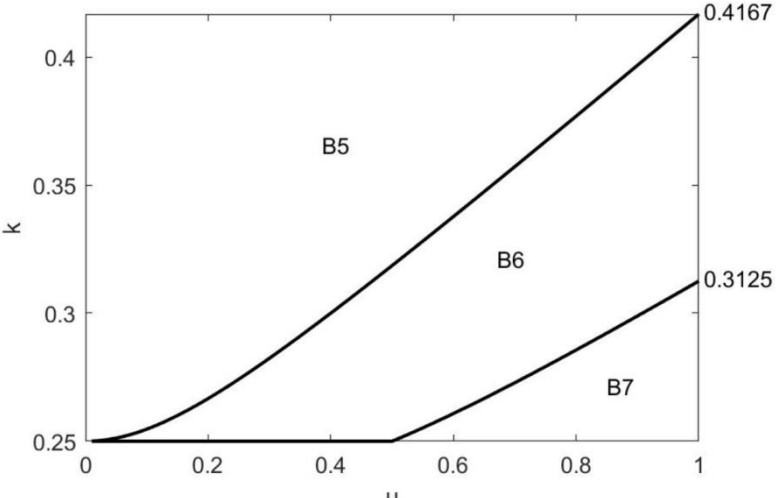

**Figure 7.** Diagram of the relationship between $\Pi_{LSSC_1}^{CC}, \Pi_{LSSC_1}^{DC}$ and $\Pi_{LSSC_2}^{CC}, \Pi_{LSSC_2}^{DC}$.

**Corollary 3**. *When (k, u) is in a particular range, for two LSSCs in which horizontal Nash competition occurs at the same level, one LSSC chooses to integrate and the other LSSC chooses not to integrate, which not only enhances its overall profit but also promotes the overall profit of its rival LSSC.*

**Corollary 4**. *When one LSSC chooses not to integrate, the other LSSC chooses not to integrate, thereby enhancing the profits of either its chain or one of the rival chains if horizontal Nash competition prevails at the same level.*

**Corollary 5**. *In two LSSCs where horizontal Nash competition occurs at the same level when (k, u) is in a certain range, two chains choosing opposing integration strategies will not only promote the overall profit of their rival LSSC but also increase their overall profit. Two chains choosing the same strategy will, at most, increase the profit of either their chain or their rival chains.*

Similar to the reasons stated in the preceding section, the service efficiency investment factor also limits the impact of service efficiency competition intensity on supply chain integration decisions and profits.

## 6. Conclusions and Prospects

The findings of the study and the management implications are as follows.

### 6.1. Conclusions

(1) When the service efficiency investment coefficient is extremely large (greater than a specific range of values), the integration decision of the LSSC in two logistics service supply chains in which horizontal Nash competition occurs at the same level depends only on the size of the service efficiency investment coefficient independent of the intensity of service efficiency competition, and the profit of its chain increases or decreases in the opposite direction to that of the rival chain. The supply chain can maximize its own profit by adopting the integration strategy, which reduces the profit of the rival chain at this time.

(2) The impact of two service efficiency competition intensities on the integration decision and profit of LSSCs is limited by the service efficiency investment coefficient.

① The chain's earnings rise when the competitor chain takes the integration decision when the service efficiency investment factor is large, while the competitor chain's profits fall at the same time. ② When the service efficiency investment factor is significant when the rival chain decides to integrate, its chain decides to not integrate, which might increase the profit of the rival chain while simultaneously decreasing the profit of its chain. ③ When the service efficiency investment factor is large, the competitor chain makes a non-integration decision to improve its own chain's profit while simultaneously lowering the profit of the rival chain. ④ When the service efficiency investment factor is large and significant, the rival chain's non-integration decision can result in a gain in profit for the rival chain at the expense of the own chain's integration decision.

(3) When the service efficiency improvement coefficient is small, the two service efficiency competition intensities have an impact on the integration decision of LSSC; when the service efficiency improvement coefficient is significant, the integration decision of LSSC is independent of the service efficiency competition intensity. ① From the perspective of the LSI, the non-integration strategy adopted by its chain under the lower competition intensity not only increases the profit of its chain but also increases the profit of the rival chain when the service efficiency investment factor is low and the rival chain decides to integrate. ② When the service efficiency investment factor is small and the counterparty chain adopts the decision of non-integration, from the standpoint of supply chain profitability, its chain adopts the strategy of integration under the larger competition intensity can improve its chain profit, but at the same time reduce the profit of the counterparty chain. ③ From the perspective of the LSI, the non-integration strategy adopted by its chain under less competitive intensity not only increases its chain profit but also makes the rival chain profit higher when the service efficiency investment factor is low and the rival chain decides against integration. ④ When the service efficiency investment factor is small and the counterparty chain adopts the decision of non-integration, from the perspective of supply chain profitability, its chain adopts the integration strategy under the larger competition intensity and not only improves its chain profit but also increases the profit of the counterparty chain.

(4) The effect of competition intensity on the LSSC decision is limited by the service efficiency investment coefficient. When the service efficiency investment coefficient is constant, the optimal decision of the supply chain changes from non-integration to integration as the competition intensity increases.

(5) In the process of competition between two LSSCs, the LSP service efficiency investment factor and the intensity of service efficiency competition influence the integration decisions of each LSSC, and the indicators such as service efficiency investment factor and service efficiency competition intensity cause changes in the vertical integration decisions of the rival chain when choosing an integration or non-integration strategy.

*6.2. Managerial Implications*

(1) When the cost required to improve the efficiency of unit services is particularly large, adopting an integration strategy to achieve rational use of expendable resources can help achieve the goal of sustainable development and is an effective measure to increase profits at the same time.

(2) When a supply chain adopts an integration strategy, the manager adopts an integration strategy to achieve the rational use of consumable resources, which helps to achieve the goal of sustainable development and is an effective measure to increase the profit of the chain. When a chain adopts the non-integration strategy, the manager adopts the integration strategy to achieve the rational use of consumable resources, which helps to achieve the goal of sustainable development, but this measure is ineffective to increase the profit of its own chain.

(3) When the cost of improving the efficiency of unit services is small, managers' adoption of integration strategies to achieve rational use of expendable resources contributes to

the goal of sustainable development and is an effective measure to increase the profitability of their own chains.

(4) When improving the efficiency of unit services requires fixed costs, the greater the intensity of competition between chains, the more effective it is for managers to adopt integration strategies to achieve rational use of expendable resources and make their own profits increase.

(5) Choosing a vertical structure requires consideration of both its future development strategy and the market's horizontal competitive forces. The business can maximize its profitability if internal and external elements are properly balanced. It can only improve its competitiveness by promptly adapting its vertical strategy to the various market circumstances and by prudently avoiding risks.

*6.3. Prospects*

However, there are some limitations to this study. First, it is possible that two LSSCs play a master–slave game and that the service efficiency investment coefficients of two LSPs might not be equal. However, this paper only considers the case of equal rights between the two chains and equal service efficiency coefficients. Second, to ensure that the supply chain's overall profit is optimal, this study examined the best integration decision for the supply chain from the perspective of supply chain profit. However, determining how to allocate the supply chain's increased profit after this decision will require additional research. Third, this study only examines the impact of a single dimension of service efficiency on the decision to integrate a competitive supply chain, while there may be multiple dimensions of service evaluation such as service efficiency and goods security that are not mutually exclusive, and the impact of multiple dimensions together on the decision to integrate a competitive supply chain will require further research in the future.

**Author Contributions:** Conceptualization, X.Z. and Q.Z.; methodology, X.Z. and J.Z.; software, X.Z. and X.Y.; validation, X.Z., Q.Z., J.Z. and X.Y.; formal analysis, X.Z., Q.Z. and J.Z.; investigation, X.Z., Q.Z., J.Z. and X.Y.; writing—original draft preparation, X.Z. and J.Z.; writing—review and editing, X.Z., Q.Z., J.Z. and X.Y. All authors have read and agreed to the published version of the manuscript.

**Funding:** This research was supported by the National Natural Science Foundation of China (NSFC) under grant number 72163025, the Humanities and Social Science Research Program of Ministry of Education under grant number 20YJA630090, the Inner Mongolia Autonomous Region Youth Science and Technology Talents Support Program under grant number NJYT22035, and the Natural Science Foundation of Inner Mongolia Autonomous Region under grant number 2021MS07022.

**Institutional Review Board Statement:** Not applicable.

**Informed Consent Statement:** Not applicable.

**Data Availability Statement:** Not applicable.

**Acknowledgments:** The authors are grateful to the anonymous reviewers and editorial team for their constructive comments on the study.

**Conflicts of Interest:** The authors declare no conflict of interest.

**Appendix A**

The process of proving Lemma 1.

$$\Pi_{LSSC1}^{CD} - \Pi_{LSSC1}^{DD} = \frac{k(4k-1)(8k-1-u)^2(a-c_{LSI}-c_{LSP})^2}{(1+32k^2-12k-u^2)^2} - \frac{k(12k-1)(a-c_{LSI}-c_{LSP})^2}{(8k+u-1)^2}$$

$$= \frac{8k^2(a-c_{LSI}-c_{LSP})^2}{(8k+u-1)^2(1+32k^2-12k-u^2)^2} * (-256k^3 + 512k^4 + u^2 - u^4 + 8k^2(5+4u^2) - 2k(1+6u^2))$$

where $\sigma_1 = (a - c_{LSI} - c_{LSP})^2$

$$\Pi_{LSSC1}^{CD} - \Pi_{LSSC1}^{DD} = f_1(k,u) * g_1$$

where $g_1 = \frac{8k^2\sigma_1}{(8k+u-1)^2(1+32k^2-12k-u^2)^2} > 0$

$$f_1(k,u) = -256k^3 + 512k^4 + u^2 - u^4 + 8k^2(5+4u^2) - 2k(1+6u^2)$$

So let $f_1(k,u) = 0$, which gives the undifferentiated curve $u = u_1(k)$.

As shown in Figure 2, when $0.25 < k < 0.3259$, there is a monotonic progression of $u_1(k) \in (0,1)$, $u_1(k)$ about $k$.

On the upper-left side of the undifferentiated curve $u = u_1(k)$, i.e., $0.25 < k < 0.3259$ and $0 < u < u_1(k)$, there is $f_1(k,u) > 0$, then $\Pi_{LSSC_1}^{CD} > \Pi_{LSSC_1}^{DD}$.

On its lower-right side, i.e., $0.25 < k < 0.3259$ and $u_1(k) < u < 1$ has $f_1(k,u) < 0$, then $\Pi_{LSSC_1}^{CD} < \Pi_{LSSC_1}^{DD}$.

When $k \geq 0.3259$, $f_1(k,u)$ is monotonic about $u$, $f_1(k,0) = 512k^4 - 256k^3 + 40k^2 - 2k > 0$, and $f_1(k,1) = 512k^4 - 256k^3 + 72k^2 - 14k > 0$,

We have established that when $k \geq 0.3259$ and $0 < u < 1$ $f_1(k,u) > 0$, $\Pi_{LSSC1}^{CD} > \Pi_{LSSC1}^{DD}$. Thus, Lemma 1 is proved.

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
