# Peer review of "Logistics Service Supply Chain Vertical Integration Decisions under Service Efficiency Competition"

_sustainability, doi:10.3390/su15053915_

Round 1

Reviewer 1 Report

The authors presented an interesting vertical integration scheme for logistics service supply chain. Inverse derivation and comparative analysis were used to examine the relationship between profit and efficiency of supply chain with and without integration strategy.

There are some aspects of the article that need to be improved as described below.

 1. Introduction: The sound of sustainability in this paper is not enough. To keep this article in the scope of the journal, authors should clearly state the sustainability contributions of their research paper. The research gap (in the sense of sustainability) should obviously be stated in the article, both theoretical gap and theoretical necessity. How the vertical integration effect to the sustainability? The scope of journal can be found in https://www.mdpi.com/journal/sustainability/about and the extension of sustainability to the SDGs is also found at https://www.undp.org/sustainable-development-goals

2. Literature review: As a result of journal name and scope, your article needs to be improved the sense of sustainability. This could be done by adding review and discussion related to “sustainability”. I recommended more than 10 references in sustainable supply chain, green supply chain, circular supply chain, reverse logistics, and other related topics need to be addressed. However, superficial citation is not acceptable. Authors have to clearly state the impact or relevance of sustainability issues with supply chain profitability, supply chain competition, supply chain efficiency, or integration strategy. Moreover, the word “thesis” in paragraph 3 of section 2.1 should be changed.

3. Discussion: This article does not have an active "discussion section” to show the theoretical contributions of this research to the scientific literatures. The comparisons must include confirmatory findings (similarity of each findings) or counterintuitive findings (dissimilarity) to the results of published articles. Authors should argue how their research stands among other published papers within the body of knowledge. The active discussion section needs to be organized into subsections according to the research questions. If possible, one specific research question should be referred in each subsection. Readers will be convinced that the article’s objectives were totally fulfilled if the research findings are clearly answer research questions.

Reviewer 2 Report

The authors aim at studying the relationship between the overall profit of a chain and that is a rival chain under the service efficiency competition with and without integration strategy.

Although authors argue that competition is nowadays based on service quality, the focus of the research is on efficiency. It is not clear what is the role of efficiency in the research and why other factors, such as service quality, are not included.

Two RQs are offered. These are holistic and the method used in the research does not allow providing an appropriate answer to these RQ.

Through the introduction, there are several statements that show the personal opinion of the authors and do not translate the state of the art in SC knowledge. As an example: “we are all aware that having a centralized supply chain structure is necessary to create an ideal system.”. This shows a clear lack of regard for the latest research regarding supply chain management.

Part of the LR is dedicated to a topic that the authors themselves conclude is not the focus of the research. If it does not add value to the research, it should not be included in the article.

It is not clear why the methods used were selected. Support for this adoption should be provided.

In the problem description, the authors refer to “customer experience”. Then develop a model based on cost and profit. Current knowledge in SC assessing “customer experience” beyond the comparison of cost and profit.

The conclusion focuses on optimization models, but there is no effective answer to the RQs raised.

I was expecting to find a discussion of the results in comparison with the existing literature but did not find it.

Other issues:

-          The English writing is very strange and required revision. There are sentences that make no sense. Example: “Vertical integration has evolved from vertical integration (…)” (1st sentence of the introduction)

-          Acronyms should have a description of their meaning the first time they are used

-          The referencing system should be adjusted to the journal standard

-          Although the research might have been conducted under the scope of a thesis, it is now being presented as a scientific article. Authors should not refer to it as a thesis.

Reviewer 3 Report

Thanks for submitting your paper to Sustainability. This paper aims to the relationship between the overall profit of its chain and that of the rival chain under the service efficiency competition with or without the integration strategy by using inverse derivation method and comparative analysis. Overall, the paper is well written, and flowed logic looks robust. Please see my comments below.

-      In the introduction, first sentence says that “Vertical integration has evolved from vertical integration”. I wonder whether it is right expression or not.

-      In the intro, the expression “port enterprises” can be changed “port (terminal) operators”.

-      Please explain why the port enterprises can be seen as logistics service integrators in detail. In general, large shipping companies try to integrate their service from shipping towards road, rail for door-to-door services.

-      In the intro, I can see some cooperation examples in maritime industries, but some reviews from the perspective of academic journal should be reflected there. Please reflect some academic journal papers regarding cooperation in the maritime industries (e.g. (1) The influence of supply chain collaboration on collaborative advantage and port performance in maritime logistics, International Journal of Logistics: Research and Applications 19 (6), 562-582, (2) Measures of supply chain collaboration in container logistics, Maritime Economics & Logistics 17 (3), 292-314)

-      I think that research objectives are quite clear, but I’d like to see clearer research gaps prior to the research objective. Please highlight why the current study is unique and important with convincing arguments.

-      Methodology and result parts look well-written and robust, but conclusion parts need to be improved. The current conclusion only has main results and limitation. It should have some decent theoretical or managerial implications derived from the results. Also, the results can be compared to previous studies’ ones if similar prior studies exist.  

Reviewer 4 Report

Brief summary

I received an exciting paper for review, which focuses on service efficiency-sensitive customers and examines supply chain strategies from an efficiency perspective. There is no doubt that the paper presented here makes a scientific contribution to the field of supply chains and logistics. The main interest of the paper is in the integration of decisions on the profits of the self-chain and competitive chain, but how SDG goals are approached within these strategies remains unclear. And this is where my only and biggest concern is. The paper is not suitable for the journal Sustainability.

I suggest authors and editors publish it in the field of mathematics (maybe MDPI Mathematics journal), supply chains or logistics. I come from the forestry sector, where we are perhaps a little more sensitive to the term 'sustainable' and 'sustainability', and we feel that the principle has been applied too often in too general a way in many areas recently. The paper also gives the impression that the 'sustainability' aspects (including the SDGs) are somehow shoehorned into the paper to fit the aims and scope of the Sustainability journal. The authors should demonstrate the sustainability aspects of the study in the results and the comments in the discussion because everything more efficient is not necessarily more sustainable. Efficiency is only one of the stated objectives of sustainable development.

General concept comments 

Line Numbers are missing and are therefore very hard to specify the location of comments. However, there are some minor issues that apply to the paper as a whole:

·      Check also the crossreferencing (e.g. [12]studied structural ... shall, in my opinion, be Seo et al. [7] collected ). 

·      Some spaces are missing between sentences (e.g. ... structural choice.In order....), check also double spaces.  

·      Abberation was not introduced (LSSC's integration), also some other aberrations are used in the manuscript and are not introduced. 

·      Both two research questions are numbered 1

Reviewer 5 Report

3. Problem description and underlying assumptions.

The section contains the formula for the demand for logistics services. After it, the decoding of the formula is indicated. There are inconsistencies in the decoding and formula. For example, pi and p3−i. In the formula p3−i. Absent, there is e3-i. Then the designations that are not in the formula are given. The boundary between Wang and Liu's competitive service supply chain model [5] and the authors' hypothesis has been erased.

Recommendation: write more clearly what is used from the work of Wang and Liu and the authors' hypothesis implemented in section 4.

5. Comparative profit analysis of different integration decision models

There are discrepancies in the numbers of figures and references to these figures in the text. So, in the figures end-to-end numbering is used (Fig. 2, 3, etc.). And the text indicates other numbers (5.1; 5.2; 5.3, etc.)

Round 2

Reviewer 1 Report

The authors tried to fulfill all the comments from reviewers, however, it is still inadequate.

The lacking of the sound of sustainability is the main reason for rejection this paper. Although the authors added some related articles, it was not blended into the core structure of the paper. The active discussion is also needed, but is not fulfilled.

Last, but not the least, the authors should explain more detail in the responses to reviewer. Normally, it should be stated all issues changed and their line numbers.

Reviewer 2 Report

Authors state that, according to their opinion, their customers can be divided into those sensitive to service efficiency, those sensitive to cargo security, others sensitive to billing accuracy, etc. The article focuses on those that are sensitive to service efficiency. Then the authors provide an analysis regarding the impact of integration decisions on competition based on efficiency. As the groups of customers might not be mutually exclusive, the impact of the focus on efficiency on the other dimensions should be considered.

 Although the authors reinforced the conclusions of the article, it still missing a discussion in which the findings of this research are compared with findings from other articles already published.

 Personal opinions should be removed from the article. As a scientific document, the ideas should be grounded in literature and be aware that there can be contradicting opinions.

 Although the article includes rewritten parts, it does not evolve much in terms of clarity. There is an evolution, but it is still difficult to read and two of the main reasons for it are: the lack of flow in the ideas presented and the poor use of the English language. The authors mentioned that the article when through revision by a native English editor, but it does not show much impact on the text. Eventually, the document continued to receive adjustments even after that revision, but the result is a document that is not easy to read.

Reviewer 3 Report

Thanks for revising your paper. I think that paper's quality is much enhanced, so I recommend 'accept'. 

Round 3

Reviewer 1 Report

After reconsideration, this article can be accepted without any further comments.

Reviewer 2 Report

The article still does not have a substantial discussion of its results. Although one reference was included in the conclusions, it is still very modest to be called a discussion.

Shortcomings of the research, such as not considering other service characteristics customers might value, are now identified as future research. These should not be considered as topics for future research, but instead included in the current article. Not considering these several dimensions leads the research to be very short in terms of its contribution.

The English writing should be revised by a professional reviewer that is at the same time a native speaker.

Round 4

Reviewer 2 Report

The authors made an effort to include implications for practie, which is highly appreciated.